# Discordance between Invasive and Non-Invasive Coronary Angiography: An In-Depth Functional and Anatomical Analysis

**DOI:** 10.3390/biomedicines11030913

**Published:** 2023-03-15

**Authors:** Shigetaka Kageyama, Kaoru Tanaka, Shinichiro Masuda, Momoko Kageyama, Scot Garg, Adam Updegrove, Johan De Mey, Mark La Meir, Yoshinobu Onuma, Patrick W. Serruys

**Affiliations:** 1Department of Cardiology, National University of Ireland Galway, H91 TK33 Galway, Ireland; 2Department of Radiology, University Hospital Brussels, 1090 Brussels, Belgium; 3Department of Cardiology, Royal Blackburn Hospital, Blackburn BB2 3HH, UK; 4HeartFlow, Inc., Mountain View, CA 94041, USA; 5Department of Radiology, Universitair Ziekenhuis Brussel, VUB, 1090 Brussels, Belgium; 6Department of Cardiac Surgery, Universitair Ziekenhuis Brussel, VUB, 1090 Brussels, Belgium; 7Imperial College London, London SW7 2BU, UK

**Keywords:** CABG, coronary computed tomography angiography (CCTA), fractional flow reserve derived from CCTA (FFR_CT_), non-invasive coronary angiography, invasive coronary angiography

## Abstract

A 79-year-old male with chronic coronary syndrome with complex coronary artery disease was included in the first-in-man trial of surgical revascularization guided solely by coronary computed tomography angiography (CCTA) and fractional flow reserve derived from CCTA (FFR_CT_). In CCTA analysis, the patient had calcified three-vessel disease, with a global anatomical SYNTAX score of 27. In contrast, in the initial FFR_CT_, only the ramus intermediate stenosis was physiologically significant, with no other vessels having an FFR_CT_ ≤ 0.80 (functional SYNTAX score of 2). Discordance between the results of the CCTA and FFR_CT_ necessitated an in-depth analysis by using both invasive and non-invasive coronary angiography. Angiography-derived fractional flow reserve (FFR) confirmed that the stenosis in the proximal left anterior descending artery (LAD) was physiologically significant, while it remained functionally negative in the second assessment of FFR_CT_. Extensive calcification is the most plausible explanation for the underestimation of the stenosis of proximal LAD in CCTA-derived FFR technology.

## 1. Patient Characteristics and the Study Protocol

A 79-year-old male with chronic coronary syndrome and three-vessel coronary artery disease (CAD) on the background of smoking and non-insulin-dependent diabetes was enrolled in the FASTTRACK CABG trial, a first-in-man study assessing the safety and feasibility of planning and executing surgical revascularization in patients with complex CAD based solely on coronary computed tomography angiography (CCTA) combined with fractional flow reserve derived from CCTA (FFR_CT_) [1].

In more detail, the trial is an investigator-initiated single-arm, multicenter, prospective, proof-of-concept, first-in-man study exploring the safety and feasibility of planning and performing coronary artery bypass grafting (CABG) with the sole guidance of CCTA and FFR_CT_ (HeartFlow, Inc., Mountain View, CA, USA) in patients with the three-vessel disease (3VD) with or without the left main disease (NCT0414202). In this trial, surgical revascularization is executed by a Heart Team consisting of a cardiac surgeon and a cardiac radiologist who have no access to conventional invasive coronary angiography (ICA). The feasibility assessment is defined by the formal request of the CTA/surgical team to be unblinded to the findings of ICA, considering their reluctance to perform surgery under the sole guidance of CCTA and FFR_CT_. The safety assessment of this novel diagnostic approach relies on CCTAs performed 30 days after CABG, to assess graft patency with the topographical revascularization, when comparing the surgical planning. For this first-in-man trial, virtual multicenter Heart Team discussions are held every week.

In the FASTTRACK CABG trial, written informed consent was taken by each participating center, and the study was approved by the institutional review board (2020-1889_1-BO).

## 2. Study Procedures and Results

The CCTA was analyzed centrally by a core laboratory. Volume rendering assessment identified severe calcification in the proximal left anterior descending artery (LAD, Figure 1A). In the right coronary artery, there was a focal stenosis in the posterior descending artery (PDA, Figure 1B). In the left coronary artery, there was a severely calcified diffuse lesion involved in the bifurcation of the proximal LAD with the first diagonal (Figure 1C) and isolated stenoses in the proximal left circumflex (LCX) and ramus intermediate (RI, Figure 1D). In summary, the patient had calcified three-vessel disease (3VD) with a global anatomical SYNTAX score of 27.

In contrast, in the initial FFR_CT_, only the RI stenosis was physiologically significant (FFR_CT_ 0.70), with no other vessels having an FFR_CT_ ≤ 0.80 (Figure 2, Table 1).

Notably, a myocardial bridge was suspected in the middle of the LAD, given the vessel’s covering by myocardial tissue and its appearance when straightened (Figure 3).

## 3. Proposed Treatment by the Heart Team Versus Actual Treatment

As per the protocol, the Heart Team had the option to unblind the ICA of the patient; however, neither the Heart Team nor the surgeon requested ICA checking for this patient. The patient was assessed for the feasibility of CABG by using the SYNTAX score II and SYNTAX score II 2020, considering anatomical SYNTAX score and patient characteristics. According to the SYNTAX score II, the estimated 4 years mortality was 10.5% if the patient was treated by CABG, while 8.0% if assigned to PCI. The treatment recommendation is either PCI or CABG. SYNTAX score II 2020 predicted the 5-year major adverse cardiac event as 32.2% if treated by CABG and 45.4% if treated by PCI.

Relying on the CCTA, the Heart Team recommended using the left internal mammary artery (IMA) as a jump graft to segments 8 and 9 in conjunction with two saphenous vein grafts (SVG) anastomosed to segments 4, 16, and 12, 14a (Figure 4). Considering only the anatomic SYNTAX score and discarding the functional SYNTAX score, surgery was performed without requesting access to the conventional invasive coronary angiogram (ICA).

The actual treatment received was as follows: left IMA to segments 8 and 9, SVG to segment 4, and a free right IMA anastomosed distally to segment 12 and proximally to the SVG (Y configuration, Figure 4). The myocardial bridge was identified visually; however, a myectomy was not performed as the left IMA was anastomosed distally to it. The patient was event-free after surgery, and their 30-day follow-up CCTA confirmed the patency of all grafts and anastomoses (Figure 5, Appendix A).

## 4. Post-Hoc Analysis of the Core Laboratory 

The pre-procedural ICA was analyzed by automated quantitative software: Quantitative Coronary Angiography (QCA, CAAS version 8.2, Pie medical imaging, Maastricht, Netherlands) and two angiography-derived fractional flow reserve (FFR) technologies. Quantitative Flow Ratio (QFR) was analyzed offline by QAngio XA 3D/QFR^®^ imaging software (Version 2.0.60.6 Medis medical imaging system, Leiden, Netherlands), which requires two angiographic projections with angles >25 degrees apart. AngioPlus Core software with Murray-law-based QFR (μQFR) analysis (version V2, Pulse Medical, Shanghai, China) requires only one but the best projection which presents the lesion characteristics (e.g., stenosis and length), as well as conventional QCA. Both systems have excellent diagnostic performance against invasive fractional flow reserve (FFR, area under the curve 0.94 and 0.97, respectively). In each lesion, QCA and µFR were analyzed by using the same projections.

In QFR analysis, anatomic and physiologic stenosis was detected in the PDA, in a bifurcation lesion involving the proximal LAD and first diagonal, and RI; however, proximal LCX was not functionally significant (Figure 6). The same functional assessment was obtained from µFR.

QFR analysis in the systolic phase identified a QFR drop distal to the myocardial bridge, whereas the diastolic phase indicated a flow-limiting lesion in the proximal LAD (Figure 7). The QFR analysis in the systolic phase revealed a QFR drop distal to the myocardial bridge, whereas the QFR analysis in the diastolic phase indicates a flow-limiting lesion in the proximal part of the LAD.

Tissue characterization of these lesions and perivascular fat assessment in CCTA were performed by using QAngio CT RE^®^ imaging software (Version 3.2.0.13 Medis medical imaging system, Leiden, Netherlands). RI had fibro-fatty plaque and necrotic core, whereas the other plaques were mainly calcified. The perivascular fat attenuation index (FAI) did not suggest inflamed perivascular fat (Figure 8).

The contradictory findings between the ICA and the CCTA anatomy and physiology prompted re-analysis of the FFR_CT_ following remodeling of the distal PDA and distal LCX, resulting in significant drops in the FFR_CT_ in the extremities of both vessels (Figure 9).

Table 2 shows a comparison of lesion level information obtained from five different technologies: CCTA, re-analyzed FFR_CT_, QCA, QFR, and μFR. The principle of methodology to calculate the lesion parameter in FFRCT was presented in the supplement.

## 5. Discussion

Out of 79 cases enrolled in the first-in-man trial, this is the first to show discordance between CCTA and FFR_CT,_ prompting a critical reappraisal of the anatomical and functional parameters provided by the CCTA since discordance impacts decision-making, planning, and surgical execution. 

The main discussion points are:

(1) The inability of FFR_CT_ to analyze the distal part of the PDA and LAD is because vessels with a diameter <1.8 mm are not processed due to regulatory requirements of the United States Food and Drug Administration (FDA) approval [2]. At variance with this, it must be emphasized that all the significant stenoses detected by a drop in QFR on ICA were located either in the proximal or middle part of major epicardial vessels with a reference diameter as documented by QCA > 2mm.

(2) In the heavily calcified lesion, the blooming effect of calcium persisted despite changing the window level and width. The visual assessment of diameter stenosis by CCTA analysts tends to overestimate the diameter stenosis of calcified lesions [3]. Generally, sophisticated computerized analysis relying on finite elements, Hounsfield Unit subtraction, and Navier–Stokes equation performed in FFR_CT_ might be more reliable than the visual anatomic evaluation. 

(3) The reference vessel diameter was systematically smaller in CCTA than in ICA, whereas the minimum lumen diameter/area was larger, resulting in a visually underestimated diameter stenosis in CCTA. This difference mainly stems from the difficulty in delineating lumen contours in heavily calcified lesions, and this remains one of the major limitations of CCTA [4]. 

A flow-limiting lesion in the proximal LAD plays a major role in revascularization decisions [5]. Both QFR and µFR showed lesions in the proximal LAD, while no significant pressure drop was seen in the LAD on FFR_CT_ even after re-analysis, contrary to the distal RCA and LCX lesions. In contrast, both QFR and µFR presented that LAD stenosis is functionally significant. The Heart Team did not request to unblind the ICA in the case and eventually treated LAD with LIMA bypass respecting anatomically significant stenosis. Post-hoc angiography-derived FFR technologies support the decision of the Heart Team and the surgeon.

In this case, the myocardial bridge was recognized by CCTA and supported by ICA. The QFR analysis in the systolic phase of ICA revealed a QFR drop distal to the myocardial bridge, whereas the QFR analysis in the diastolic phase indicates a flow-limiting lesion in the proximal part of the LAD. On QFR, the myocardial bridge itself could be functionally flow-limiting, but that hypothesis needs to be corroborated by a dynamic test [6].

In the future, the non-invasive assessment of coronary artery disease will become more major. Feasibility and safety of the pre-procedural surgical planning of coronary revascularization solely performed by CCTA will be confirmed in the ongoing FASTTRACK CABG trial. Furthermore, PCI guidance by integration of CCTA into the catheterization laboratory is also examined to apply the clinical practice [7]. In addition, as shown in this case report, CCTA provides more information than ICA, especially concerning plaque characterization and epicardial fat analysis, which predict future events in deferred vessels. However, we also presented the limitation of CCTA-derived FFR. Moreover, low resolution compared to ICA and unstabilized image quality influenced by patient characteristics and image acquisition protocol remains the general limitation [8]. Finally, we mention that discrepancies between CCTA and other techniques can occur from time to time. For example, the discrepancy between CCTA and enhanced transthoracic echo-Doppler was previously reported to assess coronary stenosis [9].

In conclusion, we encountered a discrepancy between CCTA and FFR_CT_ assessments, prompting a re-analysis of the FFR_CT_, which resolved the discordance in the RCA and the LCX but confirmed that the proximal LAD lesion was non-flow limiting. Extensive calcification remains the most plausible explanation, although this single case exemplifies the rarity of this type of discordance in our current experience with patients having 3VD.

## Figures and Tables

**Figure 1 biomedicines-11-00913-f001:**
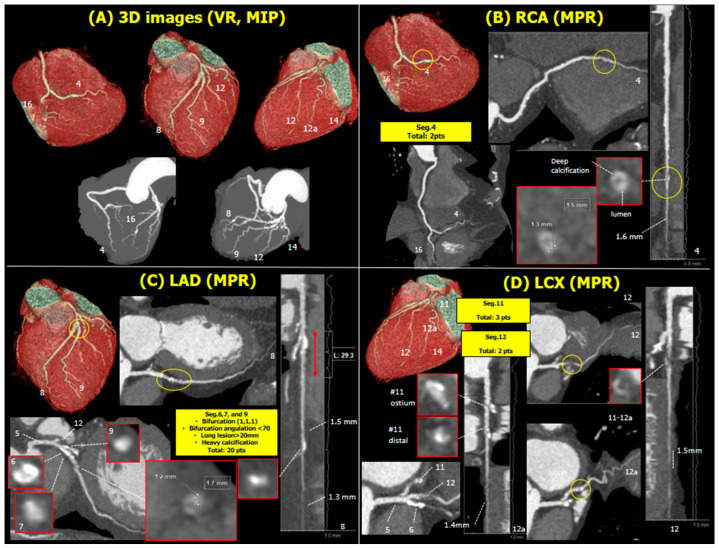
Volume rendering and maximal intensity projection with numerical identification of the 16 segments reported in the Anatomical SYNTAX score. Volume rendering (**A**), multiplanar reformation, linear projection, and cross-sectional slices of interest with the anatomic SYNTAX score points annotated (**B**–**D**). Abbreviations: 3D, three-dimensional; VR, volume rendering; MIP, maximal intensity projection; RCA, right coronary artery; MPR, multiplanar reformation; LAD, left anterior descending artery; LCX, left circumflex artery.

**Figure 2 biomedicines-11-00913-f002:**
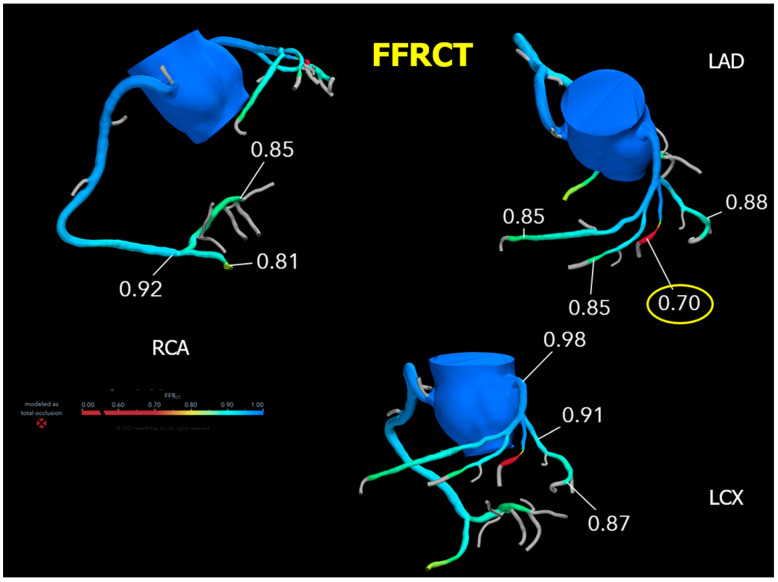
Results of the initial FFR_CT_ analysis. Abbreviations: FFRCT, fractional flow reserve derived from coronary computed tomography angiography.

**Figure 3 biomedicines-11-00913-f003:**
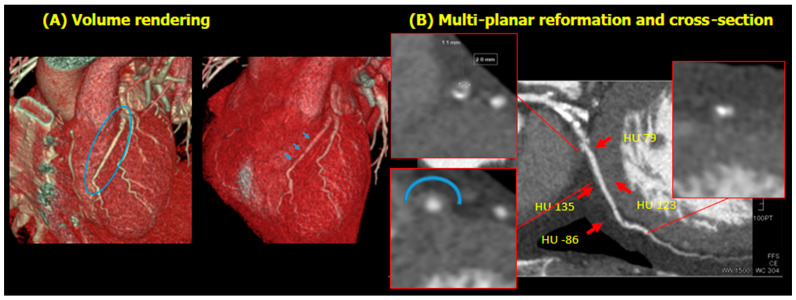
Possibility of myocardial bridge. Volume rendering image shows myocardial bridge (**A**). Unnaturally straightened LAD was presented (blue circle in the left panel). After adjusting window level, several myocardial bridges became visible (blue arrow in the right panel). Multiplanar reformation and cross-sectional image of myocardial bridge are presented (**B**). Referring to HU, the cross-section of the myocardial bridge is detectable, which is surrounded by myocardial band (blue arc in left lower cross-section). The pattern of the cross-section is different from that of organic stenosis (the other two cross-sections). Abbreviation: HU, Hounsfield unit.

**Figure 4 biomedicines-11-00913-f004:**
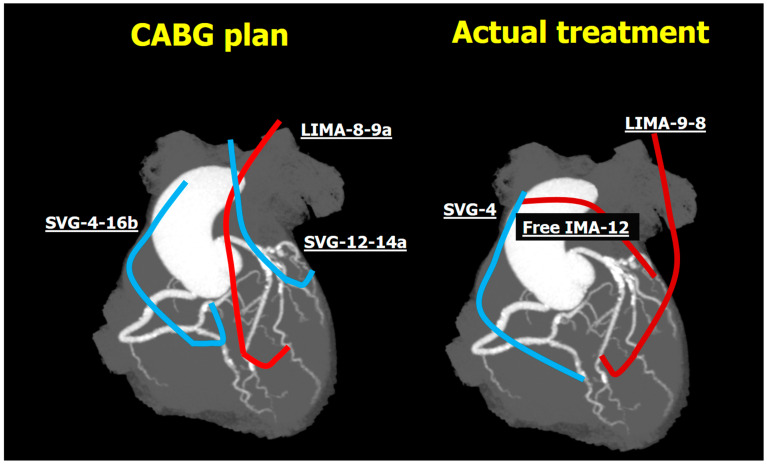
CABG plan and actual treatment. Abbreviations: CABG, coronary artery bypass grafting; (L)IMA, (left) internal mammary artery; SVG, saphenous vein grafts.

**Figure 5 biomedicines-11-00913-f005:**
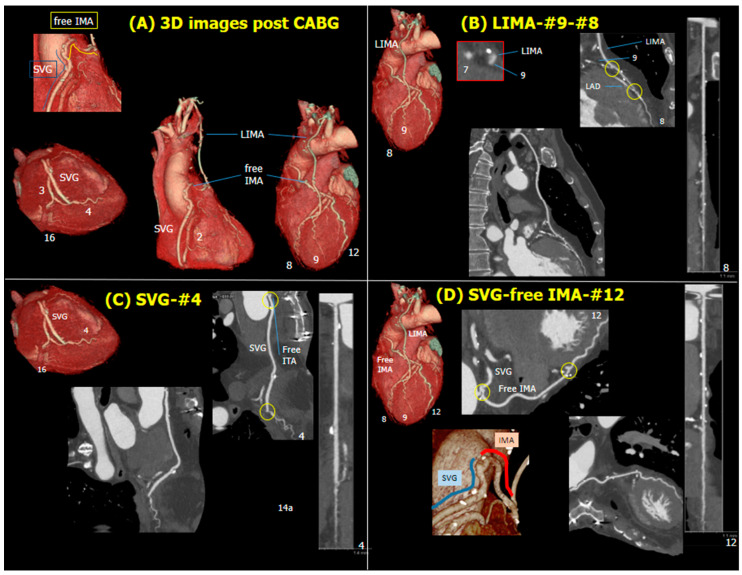
30 days follow-up computed tomography. Volume rendering (**A**), multi-planner projection and linear projection and cross-sectional slices of bypass grafts (**B**–**D**). Note that proximal anastomosis of free IMA is connected to SVG, shaping Y-graft. Abbreviations: CABG, coronary artery bypass grafting; (L)IMA, (left) internal mammary artery; SVG, saphenous vein grafts.

**Figure 6 biomedicines-11-00913-f006:**
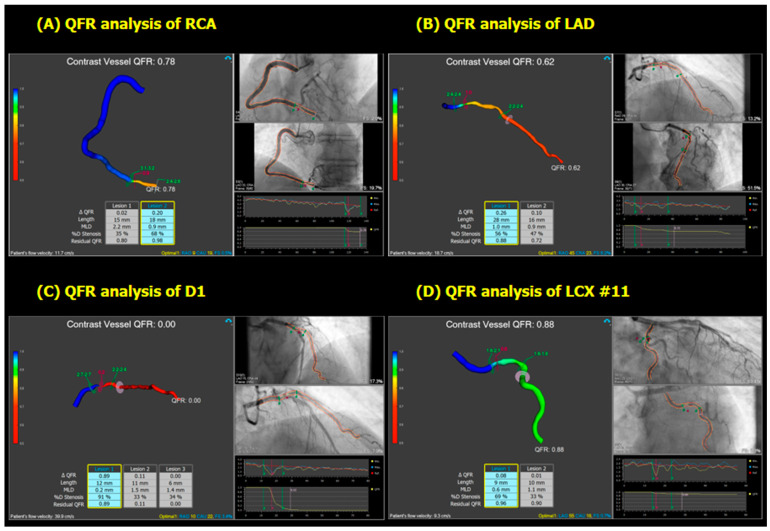
Angiography-derived FFR analysis (QFR) of the vessels containing the lesions which were detected by CCTA (**A**–**D**). Note that vessel QFR ≤0.80 is recognized as functionally significant stenosis. Abbreviations: FFR, fractional flow reserve; QFR, quantitative flow ratio; RCA, right coronary artery; CCTA, computed coronary artery angiography; LAD, left anterior descending artery; D1, first diagonal branch; LCX, left circumflex artery.

**Figure 7 biomedicines-11-00913-f007:**
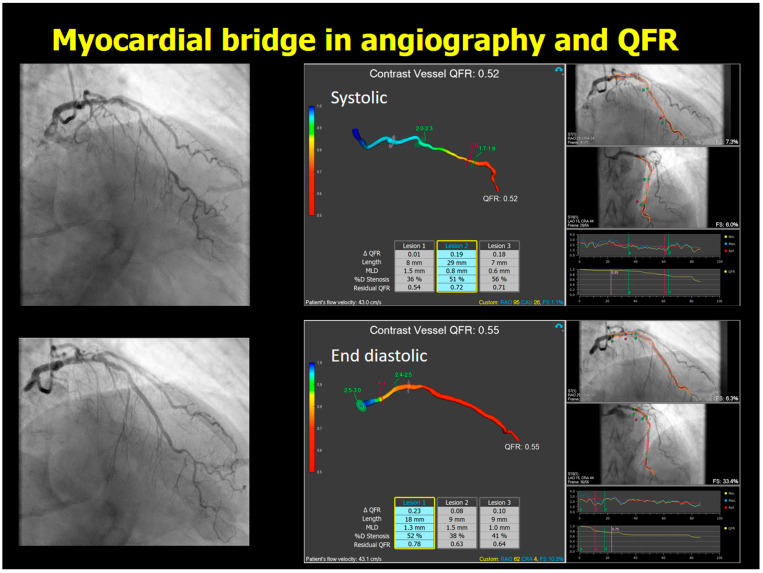
Angiography and QFR analysis of LAD, including myocardial bridge. The upper panel showed the angiography and the results of QFR analysis, and the lower panel showed those in end diastolic phase. Abbreviations: QFR, quantitative flow ratio; LAD, left anterior descending artery.

**Figure 8 biomedicines-11-00913-f008:**
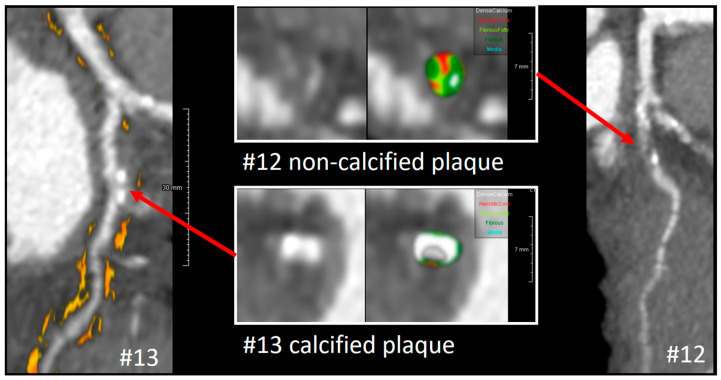
Tissue characterization and perivascular fat attenuation index in LCX. The left panel shows the multi-planner reconstruction of LCX with epicardial fat. Proximal LCX (segment 11) had calcified plaque at the point of red arrow, as presented in cross-sectional analysis with and without tissue characterization (lower center). Multi-planner of proximal intermediate artery (segment12) is presented in the right panel. A cross-section of the culprit (upper center) showed non-calcified plaque with a necrotic core. Hounsfield unit assessment of epicardial fat showed no significant inflammation. Abbreviations: LCX, left circumflex artery.

**Figure 9 biomedicines-11-00913-f009:**
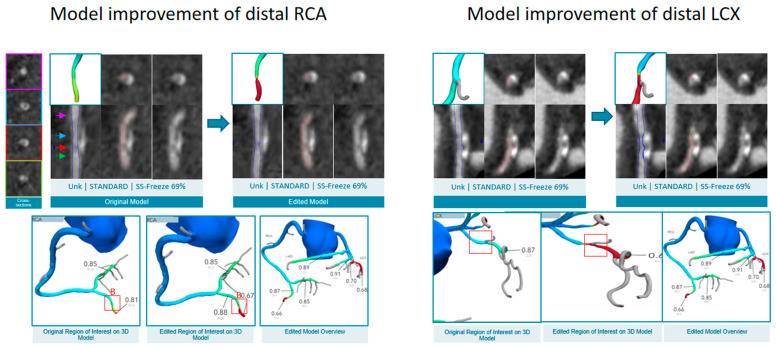
Re-analyzed FFR_CT._ The vessel contours, especially within calcified plaques, were modified, and these impacted the re-analyzed FFR_CT_ values such that the distal part of the RCA (**upper**) and LCX (**lower**) were now significant in the updated models. Abbreviations: FFRCT, fractional flow reserve derived from coronary computed tomography angiography; RCA, right coronary artery; LCX, left circumflex artery.

**Table 1 biomedicines-11-00913-t001:** Anatomical SYNTAX score and functional SYNTAX score in this case.

Lesion	Segments	Points	FFR_CT_	Final Points
1	4	2	>0.80	0
2	6, 7, 9	20	>0.80	0
3	11	3	>0.80	0
4	12	2	≦0.80	2
	**Anatomical SYNTAX score CCTA**	**27**	**Functional SYNTAX score CCTA + FFR_CT_**	**2**

Abbreviations: CCTA, coronary computed tomography angiography; FFRCT, fractional flow reserve derived from coronary computed tomography angiography.

**Table 2 biomedicines-11-00913-t002:** Obstruction analysis with 4 modalities.

		Modality					
		CCTA	Re-AnalyzedFFR_CT_	QCA from ICA		QFR	µFR
	Segment			Edge Detection	Densitometry		
**Reference diameter (mm)**	4	2.66	2.24	2.72		2.8	2.53
	6, 7	2.76	2.86	2.28		2.4	2.71
	9	2.21	1.84	2.04		2.4	1.9
	11	2.54	2.86	1.88		1.8	2.25
	12	2.77	1.8	1.79		2	1.88
**Reference area (mm^2^)**	4	5.54		5.83		6.3	
	6, 7	5.99		4.07		4.5	
	9	3.78		3.26		4.7	
	11	5.09		2.79		2.6	
	12	6.38		2.53		2.4	
**Minimal lumen diameter (mm)**	4	1.17	0.75	0.91		0.9	1.18
	6, 7	1.51	1.71	0.98		1	1.48
	9	0.57	0.83	0.23		0.2	1.05
	11	1.69	0.76	0.75		0.6	1.02
	12	0.68	0.31	0.28		0.3	0.38
**Diameter stenosis (%)**	4	58.2	60.4	66		68.2	53.2
	6, 7	48.4	18.1	57		56.5	45.3
	9	74	39.5	89		90.8	44.5
	11	35.2	58.9	60		68.8	54.5
	12	72.7	60	85		82.5	79.8
**Minimal area stenosis (mm^2^)**	4	1.07	0.52	0.65	1.14	0.9	
	6, 7	1.78	2.54	0.75	0.76	1.3	
	9	0.25	1.1	0.04	0.01	0.1	
	11	2.25	0.54	0.44	0.47	0.8	
	12	0.37	0.36	0.06	0.01	0.4	
**Area stenosis (%)**	4	82.5		89	80	85.7	78.1
	6, 7	73.4		82	81	70.7	70.1
	9	93.3		99	100	97.4	69.2
	11	58		84	83	68.9	79.3
	12	92.4		98	100	85.2	95.9
**Lesion length (mm)**	4	8	14	5.1		18	14.4
	6, 7	9	4.6	10.9		27.5	19.1
	9	16	6.1	5.3		12.4	15.1
	11	4	12.1	3.7		9.3	15.9
	12	20.8	20.3	13.9		18.7	11.4

Blue shade represents information from CCTA and red shade represents information from ICA. Abbreviations: CCTA, coronary computed tomography angiography; FFR_CT_, fractional flow reserve derived from coronary computed tomography angiography; ICA, invasive coronary angiogram; QCA, Quantitative Coronary Angiography; QFR, Quantitative Flow Ratio; µFR, single-view angiography-derived Flow Ratio.

## Data Availability

The data underlying this article will be shared on reasonable request to the corresponding author.

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
