# Peer review of "Discordance between Invasive and Non-Invasive Coronary Angiography: An In-Depth Functional and Anatomical Analysis"

_biomedicines, 2023, doi:10.3390/biomedicines11030913_

Round 1

Reviewer 1 Report

The paper presents role of  imaging in CAD.
Overall, the scientific objective is important in clinical practice. 
The article is well written and comprehensive.
The research design is appropriate and the methods clearly explained.
The interpretation of the results is clearly presented and adequately supported by the evidence adduced.
The references are up-to-date and the most important studies have been cited.
There are some minor revisions needed. Please provide a point-by-point response to the following queries.

1. Abstract is too short.

2. There is no information about bioetics cometee and patient consent.

Author Response

Thank you so much for your positive comments. Please see the attachment for point-by-point response to your comments.

Reviewer 2 Report

Thank you for asking me to review this manuscript. This is undoubtedly an interesting, well written report, highlighting  the discordance between anatomical and functional evaluation of coronary arteries, done both invasively and non-invasively.

Despite their limits, such as dependency operator, these new approaches are worthy of attention and in your analysis the external core lab lower these limits.

In literature other papers report such discrepancy between CCTA and other techniques, as E doppler TTE (can you cite them? For example Caiati et al doi 10.3390/diagnostics11020245).

Minor isssues:

In line 199 is first not fat;

In line 227 is heart and not heat.

Author Response

(The authors gave the same response as above.)
